

# Analysis of the resistance level and target site resistance mechanisms of *Echinochloa crus-galli* to penoxsulam from Hubei Province, China

Qiongnan Gu[1,2,3], Jing Shen[1,2,3], Shihai Chu[1,2,3], Qichao Huang[1,2,3], Anan Chen[1,2,3], Lin Li[1,2,3] and Ruhai Li[1,2,3]

[1] Institute of Plant Protection and Soil Fertilizer, Hubei Academy of Agricultural Sciences, Wuhan, Hubei, China

[2] Key Laboratory of Integrated Pest Management on Crops in Central China, Ministry of Agriculture and Rural Affairs, Wuhan, Hubei, China

[3] Hubei Key Laboratory of Crop Disease, Insect Pests and Weeds Control, Wuhan, Hubei, China

## ABSTRACT

*Echinochloa crus-galli* is a grass weed that infests rice fields and causes significant crop yield losses. In this study, we surveyed 15 resistant *E. crus-galli* populations collected from rice fields in Hubei Province, China, and investigated the resistance levels and target site resistance mechanisms to the acetolactate synthase (ALS) inhibitor penoxsulam. The results of whole-plant bioassay experiments revealed that 15 populations presented different levels of resistance to penoxsulam. The Trp-574-Leu mutation was detected in ten resistant populations, and the Pro-197-Leu mutation was detected in one resistant population. Additionally, the *in vitro* ALS activity in resistant populations (18-ETF, 18-WJJ, and 18-WMJ) was 51.28-, 5.51-, and 8.46-fold greater than that in the susceptible population. The ALS from these resistant populations requires a much higher penoxsulam concentration for activity inhibition. *ALS* gene expression in three resistant populations (18-ETF, 18-WJJ, and 18-WMJ) was 1.53-, 1.58-, and 1.41-fold greater than that in the susceptible population 18-NJ before penoxsulam treatment. Our results indicated that target-site mutation in *ALS* is at least partially responsible for barnyardgrass resistance to penoxsulam in Hubei Province.

## INTRODUCTION

Barnyardgrass (*Echinochloa crus-galli* (L.) Beauv.), a noxious weed commonly found in agricultural fields, gardens, and various disturbed habitats, poses a significant challenge to crop production and ecosystems. This summer annual grass species is known for its rapid growth, prolific seed production and aggressive competition, which results in decreased crop yields. Moreover, barnyardgrass has emerged as a prominent weed issue in rice-producing nations worldwide (*Rao, 2021*; *Godar & Norsworthy, 2023*; *Zhang et al., 2023*).

Corresponding author
Ruhai Li, lirh@hbaas.com

Chemical control through the use of herbicides is often necessary for the control of barnyardgrass. Herbicides commonly used for barnyardgrass control in China include penoxsulam, acetochlor, butachlor, pretilachlor, quinclorac, bispyribac-sodium, and cyhalofop-butyl (*Liu et al., 2021*). Among these, penoxsulam, a triazolopyrimidine sulfonamide herbicide, is one of the most widely utilized agents. Penoxsulam, an acetolactate synthase (ALS) inhibitor developed by Dow AgroSciences (now Corteva Agriscience), was introduced to the Chinese market in 2008 and has been extensively used owing to its effective control of barnyardgrass (*Deng et al., 2025*). However, prolonged, exclusive use of penoxsulam in recent years has resulted in the development of resistance in barnyardgrass populations across multiple regions. Notably, barnyardgrass in Hubei Province has the highest prevalence of high-level resistance to this herbicide (*Chen et al., 2016*; *Fang et al., 2019a*; *Yang et al., 2021*; *Feng et al., 2022*). Despite the widespread use of penoxsulam, the mechanisms driving *E. crus-galli* resistance in Hubei fields remain poorly characterized.

To date, barnyardgrass has developed resistance to herbicides with seven distinct modes of action, which includes acetolactate synthase (ALS) inhibitors, acetyl-CoA carboxylase (ACCase) inhibitors, photosystem-II inhibitors, auxin mimics/cellulose biosynthesis inhibitors, very-long-chain fatty acid inhibitors, microtubule assembly inhibitors, and 5-enolpyruvylshikimate-3-phosphate synthase (EPSPS) inhibitors (*Damalas & Koutroubas, 2023*). The evolution of resistance to multiple modes of action of herbicides poses a significant challenge for effective barnyardgrass control.

Herbicide resistance mechanisms in weeds can be broadly classified into two main categories: target-site resistance (TSR) and non-target-site resistance (NTSR). The TSR mechanism is associated with genetic modifications or mutations, as well as the overexpression or amplification of genes that encode the target proteins of herbicides. The NTSR mechanism involves mechanisms that facilitate the metabolism of herbicides within the plant, rendering the herbicides less effective, or that redirect the herbicide away from its intended site of action. However, among these two types of resistance, TSR is more frequently reported and often confers high levels of resistance (*Powles & Yu, 2010*; *Heap, 2014*; *Yu & Powles, 2014*; *Gaines et al., 2020*). There are many TSR cases of *E. crus-galli* caused by amino acid substitutions to penoxsulam. For example, Fang et al. reported that the Ala-122-Val and Ala-205-Gly biotypes exhibited high levels of resistance to penoxsulam, and that these two biotypes have higher ALS enzyme activities than the sensitive biotype (*Fang et al., 2019a*). A Phe-206-Leu mutation in the ALS enzyme was also found to mediate resistance to penoxsulam (*Feng et al., 2022*). The Ala-122-Asn amino acid change in the *ALS* gene results in high levels of cross-resistance to the ALS inhibitors nicosulfuron, penoxsulam, bispyribac-Na, and imazamox (*Panozzo et al., 2017*). Furthermore, the Asp-376-Ser biotype also contributes to resistance to ALS-inhibiting herbicides (*Löbmann et al., 2021*). Although NTSR can be attributed to different causes, most NTSR cases have been linked to cytochrome P450 enzymes (*Pan et al., 2022*). These enzymes play crucial roles in herbicide metabolism within plants, contributing to reduced herbicide efficacy and subsequent resistance.

Located in central China, Hubei Province plays a crucial role in rice production. The effective control of barnyardgrass through herbicide application is a key factor in ensuring the continued productivity of rice cultivation in this area. Analyzing the resistance level of barnyardgrass to penoxsulam and elucidating the resistance mechanisms will reduce resistance development and be beneficial for the long-term sustainability of rice production in this important agricultural area.

## MATERIALS & METHODS

### Plant materials

Seeds of 15 *E. crus-galli* populations were collected from rice fields in Hubei Province during the 2018 growing season (Table 1 and Fig. 1). The collection was approved by the research council of the Institution of Plant Protection and Soil Fertilizer, Hubei Academy of Agricultural Sciences, with approval number 201805. The susceptible population 18-NJ was provided by the Institute of Plant Protection, Jiangsu Academy of Agricultural Sciences. All seeds were randomly collected from mature barnyardgrass plants. At least 300 individual plants were sampled at each collection point. The plant morphology and phylogenetic relationships determined by Amplified Fragment Length Polymorphism (AFLP) fingerprinting of these populations are described in detail elsewhere. The seeds were subsequently placed in a cool, dry place at room temperature (20–22 °C) to air dry and then stored in a low-temperature (1–5 °C) and low-humidity (below 50%) seed cabinet until use. The original collected populations were used for the herbicide dose–response experiments. Seeds from the resistant populations were selected with penoxsulam in a controlled greenhouse for an additional generation (F1 generation) through self-pollination.

### Plant growth conditions and herbicide application

Prior to germination, the seeds were treated with 100 mg/L gibberellic acid solution for 24 h at room temperature (20–22 °C), followed by germination in a dark incubator at 28 °C for 48 h. The germinated seedlings at the Biologische Bundesanstalt, Bundessortenamt und CHemische Industrie (BBCH) 11–12 (1–2 leaf stage) were subsequently transplanted into plastic pots (15 cm in diameter and 12 cm in height) and covered with 0.2 cm thick growth medium. The growth medium in each pot consisted of vermiculite and an organic matter mixture at a 1:1 (w/w) ratio, with each pot accommodating 30 seedlings. The pots were subsequently transferred into a growth chamber set at a temperature of 30/25 °C, with a light/dark cycle of 14/10 h, a light intensity of 12,000 lx, and a relative humidity of 60%. Before herbicide treatment, the seedlings in each pot were thinned to 20 plants whose growth was consistent. Seedlings from each population were treated with different doses of penoxsulam at BBCH 12–13 (2–3 leaf stage) by using a sprayer chamber equipped with a TeeJet® XR8002 flat fan nozzle with a spraying volume of 450 L ha$^{-1}$ at 0.3 MPa.

### Whole-plant dose response to penoxsulam

Based on the recommended field dose of 30 g a.i. ha$^{-1}$ penoxsulam in the rice field, penoxsulam doses of 1, 2, 4, 8, 16, 32, 64, 128, 256, 512, 1,024, and 2,048 g a.i. ha$^{-1}$ were applied to the putative resistant populations, and doses of 0.25, 0.5, 1, 2, 4, 8, 16, and 32 g

**Table 1** Location information of the *E. crus-galli* populations used in this study.

| Population | Location | | |
| --- | --- | --- | --- |
| | Longitude | Latitude | Altitude (m) |
| 18-ETF | 30°16′48′N | 114°55′06′E | 7 |
| 18-DDX | 30°0′51′N | 115°02′25′E | 16 |
| 18-WLH | 29°52′11′N | 115°40′15′E | 23 |
| 18-WMJ | 30°06′11′N | 115°36′48′E | 40 |
| 18-WJJ | 30°17′24′N | 114°07′56′E | 30 |
| 18-QZW | 30°14′16′N | 112°38′23′E | 30 |
| 18-JPX | 29°59′23′N | 112°32′34′E | 14 |
| 18-JJCY | 29°57′52′N | 112°37′51′E | 12 |
| 18-YTW | 29°57′25′N | 115°16′28′E | 5 |
| 18-HDD | 30°05′06′N | 115°49′32′E | 26 |
| 18-XYCZ | 31°01′26′N | 113°46′57′E | 30 |
| 18-JWZ | 29°58′12′N | 112°42′52′E | 30 |
| 18-JZWD | 31°08′37′N | 112°30′19′E | 58 |
| 18-XYXS | 31°42′00′N | 112°09′38′E | 70 |
| 18-JJXJ | 30°04′17′N | 112°26′11′E | 30 |
| 18-NJ | 32°29′43′N | 119°08′49′E | 10 |

a.i. ha$^{-1}$ were applied to the susceptible population. After herbicide treatment, the seedlings were transferred into a greenhouse with natural light, a temperature of 30/25 °C and a relative humidity of 60%. The whole experiment was conducted during the barnyardgrass growing season, which was from April to October. The aboveground fresh biomass weight of the plants was recorded at 21 days after herbicide treatment. Four replicates were performed for each herbicide dose, and the experiments were conducted twice.

### *ALS* gene cloning and sequencing

At least 10 surviving individual plants from the different dose–response treatments of each population were collected for gene identification. Approximately 100 mg of shoot tissue was used for genomic DNA extraction. Genomic DNA extraction was conducted by using a NuClean Plant Genomic DNA Kit (CWBIO, Jiangsu, China) according to the manufacturer's instructions. A specific primer pair (*ALS*-S: CTCCTTGCCACCCTCCCC and *ALS*-R: CTGCCATCACCATCCAGGATC) was designed on the basis of the *ALS* sequences (*ALS1*: LC006058.1, *ALS2*: LC006059.1, *ALS3*: LC006061.1) (Fig. S1). The polymerase chain reaction (PCR) amplification was performed with two μl of DNA template, two μl (10 μM) of each primer, 25 μl of 2×Phanta Max Buffer (Vazyme Biotech, Nanjing, China), 0.5 μl of dNTP Mix, one μl of Phanta Max Super-Fidelity DNA Polymerase (Vazyme Biotech, Nanjing, China), and ddH$_2$O to a final volume of 50 μl. The PCR program was set as follows: one cycle of 94 °C DNA denaturation for 3 min; 35 cycles of denaturation at 94 °C for 60 s, annealing at 63 °C for 50 s, and elongation at 72 °C for 120 s; and a final 72 °C DNA elongation for 10 min. The 2,000 bp PCR products were purified using a Cycle-Pure Kit (Omega Bio-Tek, Guangzhou, China) and cloned and inserted into a pTOPO vector (Genesand Biotech, Beijing, China).
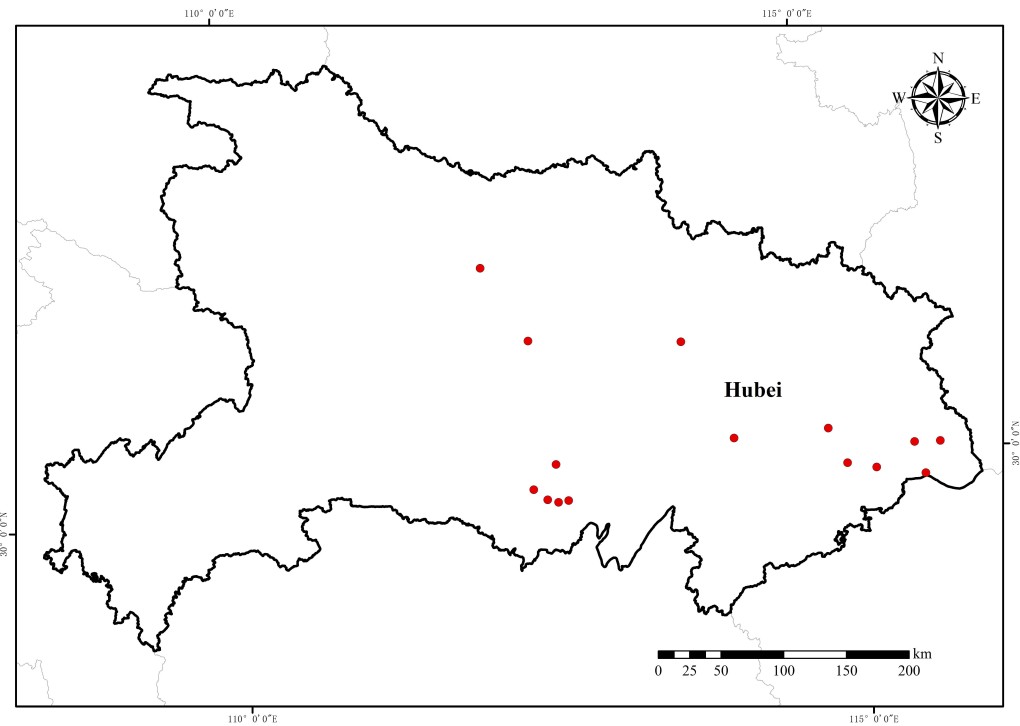

**Figure 1** **Sampling location of *E. crus-galli* resistant populations from Hubei Province collected in 2018.** Map data source: Resource and Environment Science Databases of the Chinese Academy of Science (https://www.resdc.cn/Default.aspx).

The plasmids containing the fragment insertions were transformed into DH5α chemically competent cells and sequenced by Tsingke Biotechnology (Wuhan, China). At least 10 transformed clones of each plant were sequenced to obtain *ALS* gene sequences. All the sequences were then compared *via* DNAMAN v.6 software (Lynnon Biosoft, Quebec, Canada) and the Basic Local Alignment Search Tool (BLAST) with the NCBI database to verify the gene copy and gene mutation sites.

### *In Vitro* ALS enzyme activity assay

The S populations (18-NJ) and F1 generation surviving plants of three R populations (18-ETF, 18-WMJ, and 18-WJJ) were used for further ALS enzyme activity assays. Twenty seedlings from each population were transplanted into pots and maintained in a greenhouse until they reached the three-leaf stage. Three grams of leaf tissue from each population was freshly harvested, snap-frozen in liquid nitrogen, and stored at −80 °C until use.

The protocol for the ALS enzyme extraction and assay followed the procedure of *Yu et al. (2010)* with modifications. The frozen tissue was homogenized with a mortar and pestle and suspended in four volumes of extraction buffer (one mM sodium pyruvate, 0.1 M potassium phosphate ($KH_2PO4/K_2HPO_4$), 0.5 mM $MgCl_2$, 10 µM flavin adenine dinucleotide (FAD), 100 mL/L glycerol, and 0.5 mM thiamine pyrophosphate (TPP) at pH 7.5). The mixtures were stirred for 5 min, and the homogenate was filtered through

one layer of Miracloth into Polyallomer centrifuge tubes (Beckman Coulter, USA). The samples were subsequently centrifuged for 20 min at 4 °C at 27,000× g. The supernatants were transferred to beakers, and ammonium sulfate [$(NH_4)_2SO_4$] was added at 0.313 g/mL supernatant. The supernatant and ammonium sulfate mixtures were stirred at low speed for 20 min to allow protein precipitation. The mixtures were again centrifuged at 4 °C for 20 min at 27,000× g. The supernatants were discarded, and the pellet was resuspended in 0.5 mL of extraction buffer and desalted on a Sephadex G25 column (Sigma–Aldrich, USA). Thirty-five microliters of the desalted enzyme extract was immediately added to 35 µl of the assay buffer (0.17 M sodium pyruvate, 83.4 mM potassium phosphate ($KH_2PO_4$/$K_2HPO_4$), 16.7 mM $MgCl_2$, 1.7 mM TPP, and 0.167 mM FAD at a pH of 7) and seven µl of a range of penoxsulam concentrations (0.01, 0.1, 1, 10, 100, and 1,000 µmol/L). The reactions were incubated at 37 °C in the dark for 60 min. The reaction was terminated by adding 17.5 µL of 6 N $H_2SO_4$, followed by incubation at 60 °C for 15 min. Subsequently, 87.5 µL of creatine solution (0.55% w/v) and 87.5 µL of α-naphthol solution (5.5% w/v in 5 N NaOH) were added, and the mixture was incubated at 60 °C for an additional 15 min. Enzyme activity, determined as acetoin formation, was measured colorimetrically at 530 nm. The Bradford method was used to measure the protein concentration in the crude extract (*Yu et al., 2004*). The experiment included three biological replicates and two technical repeats.

### *ALS* gene expression

The S populations (18-NJ) and F1 generation of three R populations (18-ETF, 18-WMJ, and 18-WJJ) were included in the analyses. Forty seedlings from each population were separately planted and maintained in a greenhouse until they reached the three-leaf stage. The plants of each population were sprayed with penoxsulam at 15 g a.i. ha$^{-1}$. Approximately 100 mg of leaf tissue from each population was collected from nontreated (NT) plants and treated (T) plants at 1, 3, 5, and 7 days after herbicide treatment.

Total RNA was extracted using a TRIzol total RNA extraction kit (Genstone Biotech, Beijing, China). cDNA was prepared from total RNA using 5× All-In-One RT master mix (Applied Biological Materials, Richmond, Canada). Quantitative reverse transcription PCR (RT–qPCR) was performed, and the results were analyzed with a Bio-Rad CFX Connect Real Time system using iTaq™ Universal SYBR Green Supermix (Bio-Rad, Hercules, CA, USA). The primer pair QALS-F:5′-ATCCGCATTGAGAACCTCC-3′ and QALS-R:5′-TCTTCTTGATTGCTGCACGT-3′ was used to amplify the *ALS* genes, and the β-actin gene was used as the endogenous reference gene with the primer pair ACT-F: 5′-CACACTGGTGTCATGGTAGG-3′ and ACT-R: 5′-AGAAAGTGTGATGCCAGAT-3′. The PCR conditions consisted of denaturation at 95 °C for 2 min, followed by 40 cycles of 95 °C for 3 s and 60 °C for 30 s. The relative quantification of each transcript was calculated following the $2^{-\Delta\Delta Ct}$ method (*Livak & Schmittgen, 2001*). All the qPCR assays were conducted with technical triplicates for each sample, and the experiment was repeated twice.

### Statistical analysis

All analyses were performed in a completely randomized design. The data were subjected to analysis of variance (ANOVA), and the mean differences were compared by the least

significant difference (LSD) test. Data from repeated experiments were pooled for analysis, and inspection of the error distributions suggested that the assumption of normality held reasonably well.

The $GR_{50}$ (dose at which there was a 50% reduction in growth biomass) and $IC_{50}$ (herbicide concentration leading to 50% inhibition of ALS activity) were subjected to a nonlinear log–logistic regression model using SigmaPlot (SigmaPlot Software, Chicago, IL, USA) version 14.0 (*Seefeldt, Jensen & Fuerst, 1995*):

$$Y = C + \frac{D - C}{1 + (x/g)^b}.$$

In the equation, C represents the lower limit, D represents the upper limit, $b$ represents the slope of the curve, and g represents the $GR_{50}$ or $IC_{50}$, where Y is the percentage of the control at herbicide dose or concentration X.

Resistance indexes (RIs) were calculated by dividing the $GR_{50}$ or $IC_{50}$ of the resistant population by that of the susceptible population.

## RESULTS

### Resistance levels to ALS inhibitors

The resistance levels of 15 *E. crus-galli* populations collected from rice fields in Hubei Province were surveyed (Fig. 2). Among the 15 populations, three populations presented high resistance levels to the ALS inhibitor penoxsulam, and the RIs of these three high-level resistant populations were all more than 100-fold greater than those of the susceptible population. The three high-level resistant populations (18-ETF, 18-DDX, and 18-WLH) were not injured by the recommended dose and exhibited more than 90% growth relative to untreated controls. The $GR_{50}$ values of these three populations were 750.75, 153.13, and 56.02 g a.i. ha$^{-1}$, respectively. Moreover, eight populations presented moderate resistance to penoxsulam. The RIs ranged from 10.09 to 50.60. The twelve populations (18-WMJ, 18-WJJ, 18-JPX, 18-QZW, 18-JJCY, 18-YTW, 18-HDD, 18-XYCZ, 18-JWZ, 19-JZWD, 18-XYXS, and 18-JJXJ) that presented moderate and low resistance levels partially survived under the recommended dose of penoxsulam and presented different physiological effects, such as curling leaves or shortened growth. The $GR_{50}$ values of these populations ranged from 2.88 to 24.37 g a.i. ha$^{-1}$ (Table 2).

### Sequence analysis of the genes encoding ALS enzymes

The *ALS* genes of ten individual plants from each of the 15 resistant populations were amplified and subsequently analyzed by the NCBI BLAST server for individual searches. The sequencing results were compared and distinguished from the full-length sequences of *ALS1, ALS2*, and *ALS3* in the susceptible population by DNAMAN. Among the 15 resistant populations, ten populations (18-ETF, 18-DDX, 18-WLH, 18-WMJ, 18-WJJ, 18-QZW, 18-JPX, 18-YTW, 18-HDD, and 18-XYCZ) included individuals harboring the known resistance mutation Trp-574-Leu. However, the mutated genes were not the same. 18-ETF and 18-DDX had mutations in *ALS3*, and 18-WLH, 18-WJJ, 18-QZW, 18-YTW, and 18-HDD had mutations in *ALS2,* which accounted for the largest proportion of

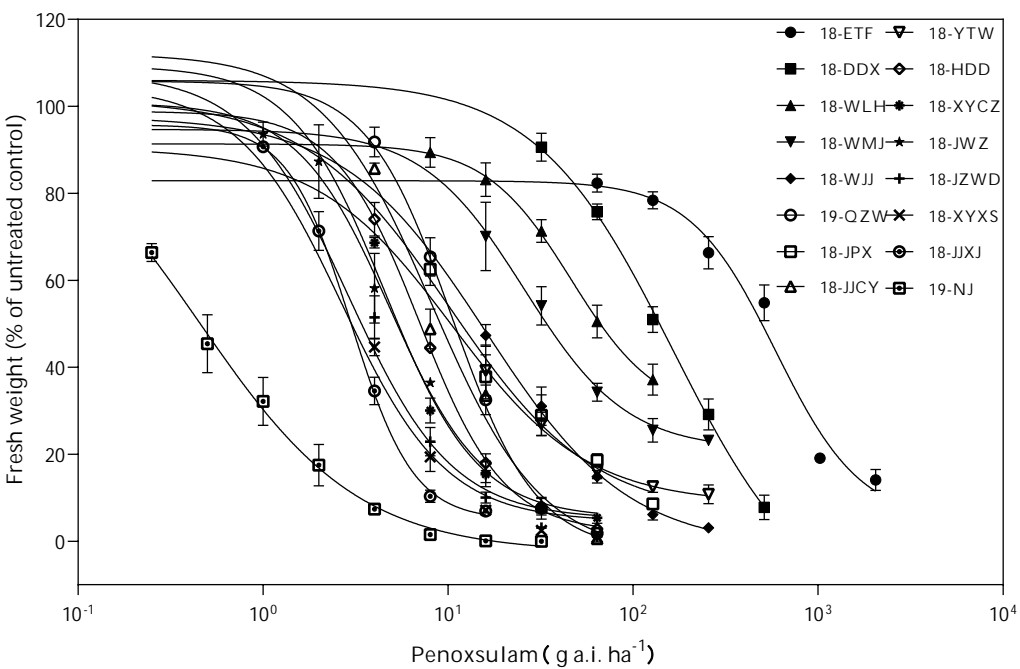

**Figure 2  Penoxsulam dose–response tests on 15 resistant and one susceptible populations.**

Trp-574-Leu *ALS* mutations, whereas 18-WMJ, 18-JPX, and 18-XYCZ had mutations in *ALS1*. Interestingly, a Pro-197-Leu substitution occurred in the 18-JJCY population. Although previous studies revealed that the Pro-197-Leu mutation is the target site basis for resistance to ALS-inhibiting herbicides in other weeds, this was the first time that the Pro-197-Leu mutation was found in *E. crus-galli*. However, none of the mutants confer resistance to known Ala-122, Ala-205, Phe-206, Asp-376, and Ser-653 sites, which have been reported in other *Echinochloa* species. In addition, four populations (18-JWZ, 18-JZWD, 18-XYXS, and 18-JJXJ) had surviving plants without any mutations in *ALS* genes, which suggested that non-target-site mechanisms may exist in these populations. More physiological and chemical analyses will be conducted to confirm this hypothesis (Table 3).

### *In Vitro* ALS enzyme activity

The F1 generations of surviving resistant plants from the 18-ETF, 18-WMJ, 18-WJJ, and 18-NJ populations were selected for further *in vitro* ALS enzyme activity assays. The total ALS activity levels measured in the resistant populations 18-ETF, 18-WMJ, and 18-WJJ were greater than those in the sensitive population, suggesting that the catalytic activity of the mutant enzyme changed (Table 4). Moreover, the $IC_{50}$ values for 18-ETF, 18-WMJ, and 18-WJJ were 63.07, 6.78, and 10.41 $\mu$M, respectively, which were 51.28-, 5.51-, and 8.46-fold greater than those of the susceptible population 18-NJ (1.23 $\mu$M) (Fig. 3). The results indicated that the Trp-574-Leu mutation could affect the conjunction of penoxsulam, which resulted in reduced sensitivity to ALS in these three *E. crus-galli*

**Table 2  Parameter values of dose–response curves to penoxsulam for *E. crus-galli* high and moderate resistant populations.**

| Population | Regression parameters | | | | $GR_{50}^{a}$ (g a.i. ha-1) | RI[b] |
|---|---|---|---|---|---|---|
| | C | D | b | $R^2$ | | |
| 18-ETF | −23.8498 | 98.0456 | 0.9278 | 0.9839 | 750.7525 | 2,668.87 |
| 18-DDX | −8.5765 | 100.5771 | 1.3927 | 0.9971 | 153.1344 | 544.38 |
| 18-WLH | 12.2321 | 99.3802 | 1.1526 | 0.9912 | 56.0235 | 199.16 |
| 18-WMJ | 19.3313 | 99.8202 | 1.4147 | 0.9920 | 24.3758 | 86.65 |
| 18-WJJ | −2.318 | 99.9599 | 1.0902 | 0.9982 | 15.5061 | 55.12 |
| 18-QZW | −1.0278 | 100.6300 | 2.1266 | 0.9977 | 11.1338 | 39.58 |
| 18-JPX | 5.0945 | 100.2026 | 1.1199 | 0.9929 | 10.7240 | 38.12 |
| 18-JJCY | −3.4679 | 101.3554 | 1.6238 | 0.9919 | 9.4823 | 33.71 |
| 18-YTW | 8.3816 | 99.9890 | 1.1374 | 0.9968 | 8.9614 | 31.86 |
| 18-HDD | 1.8565 | 99.9810 | 1.8835 | 0.9988 | 6.9076 | 24.56 |
| 18-XYCZ | 6.7458 | 100.1820 | 2.3899 | 0.9983 | 5.2352 | 18.61 |
| 18-JWZ | 6.7779 | 100.1665 | 1.7734 | 0.9890 | 4.8596 | 17.28 |
| 18-JZWD | 0.7337 | 100.0367 | 1.7694 | 0.9981 | 4.0714 | 14.47 |
| 18-XYXS | 0.4137 | 100.0031 | 1.7502 | 0.9992 | 3.5175 | 12.50 |
| 18-JJXJ | 4.1401 | 99.2003 | 2.358 | 0.9982 | 2.8776 | 10.23 |
| 18-NJ | −2.5043 | 100.0538 | −1.0260 | 0.9913 | 0.2812 | 1.00 |

Notes.
[a]$GR_{50}$, herbicide rate causing 50% growth reduction of plants.
[b]RI (resistance index) = $GR_{50}$ value of R population/$GR_{50}$ value of S population.
Resistance classes: RI≤3, susceptible; 3<RI≤10, low resistance; 10<RI≤100, moderate resistance; RI>100, high resistance.

**Table 3  Mutations identified in ALS from *E. crus-galli* resistant plants.**

| Population | No. of plants sequenced | No. of plants with mutation | Resistance mechanism | ALS mutation site | Mutation copy | Accession number |
|---|---|---|---|---|---|---|
| 18-ETF | 10 | 10 | TSR | Trp-574-Leu | ALS3 | PV393065 |
| 18-DDX | 10 | 9 | TSR | Trp-574-Leu | ALS3 | PV393064 |
| 18-WLH | 10 | 8 | TSR | Trp-574-Leu | ALS2 | PV393070 |
| 18-WMJ | 10 | 10 | TSR | Trp-574-Leu | ALS1 | PV393071 |
| 18-WJJ | 10 | 9 | TSR | Trp-574-Leu | ALS2 | PV393069 |
| 18-QZW | 10 | 1 | TSR | Trp-574-Leu | ALS2 | PV405280 |
| 18-JPX | 10 | 8 | TSR | Trp-574-Leu | ALS2 | PV393068 |
| 18-JJCY | 10 | 9 | TSR | Pro-197-Leu | ALS3 | PV393067 |
| 18-YTW | 10 | 10 | TSR | Trp-574-Leu | ALS2 | PV393073 |
| 18-HDD | 10 | 9 | TSR | Trp-574-Leu | ALS2 | PV393066 |
| 18-XYCZ | 10 | 9 | TSR | Trp-574-Leu | ALS1 | PV393072 |
| 18-JWZ | 10 | 0 | Suspect NTSR | – | – | |
| 18-JZWD | 10 | 0 | Suspect NTSR | – | – | |
| 18-XYXS | 10 | 0 | Suspect NTSR | – | – | |
| 18-JJXJ | 10 | 0 | Suspect NTSR | – | | |

Table 4 ALS-enzyme activity and IC$_{50}$ values in ALS-enzyme activity assays.

| Population | ALS activity (μM) | R/S | IC50 (μM) | R/S |
|---|---|---|---|---|
| 18-ETF | 34.73 ± 0.37 | 1.73 | 63.07 | 51.28 |
| 18-WMJ | 30.03 ± 0.69 | 1.49 | 6.78 | 5.51 |
| 18-WJJ | 32.82 ± 0.59 | 1.63 | 10.41 | 8.46 |
| 18-NJ | 20.07 ± 0.29 | – | 1.23 | – |

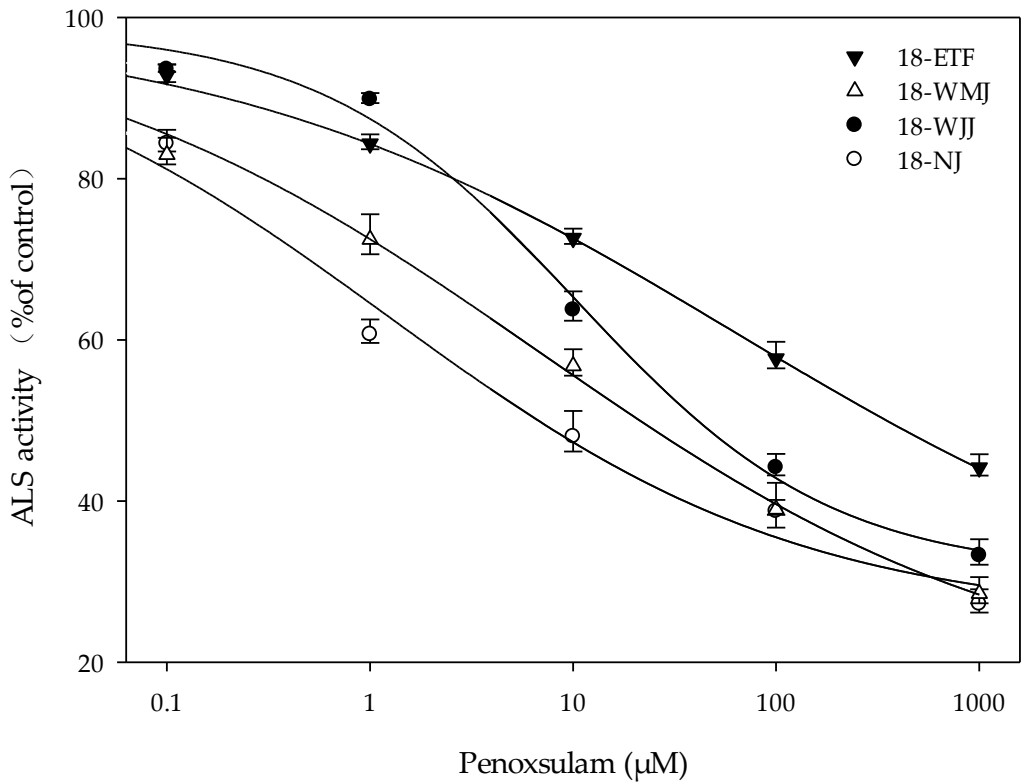

Figure 3 Response curves for *in vitro* acetolactate synthase activity of resistant (18-ETF,18-WMJ, 18-WJJ) and susceptible (18-NJ) biotypes in response to penoxsulam relative to controls. Test penoxsulam concentrations ranged from 0 to 1,000 mM and are presented on a log scale. Bars represent standard error.

populations. In addition, this trend was not completely consistent with the whole-plant dose–response analysis, in which the IC$_{50}$ value for 18-WMJ was lower than that for 18-WJJ. This may be related to different *ALS* gene mutations and mutation frequencies. Although a slight difference in *in vitro* enzyme activity was detected, the results supported the hypothesis that a TSR mechanism was responsible for the resistance of the *E. crus-galli* 18-ETF, 18-WMJ, and 18-WJJ populations.

### *ALS* gene expression

*ALS* gene expression analysis was conducted in the R populations 18-ETF, 18-WJJ, 18-WMJ and S population 18-NJ. In the absence of penoxsulam treatment, the basal expression

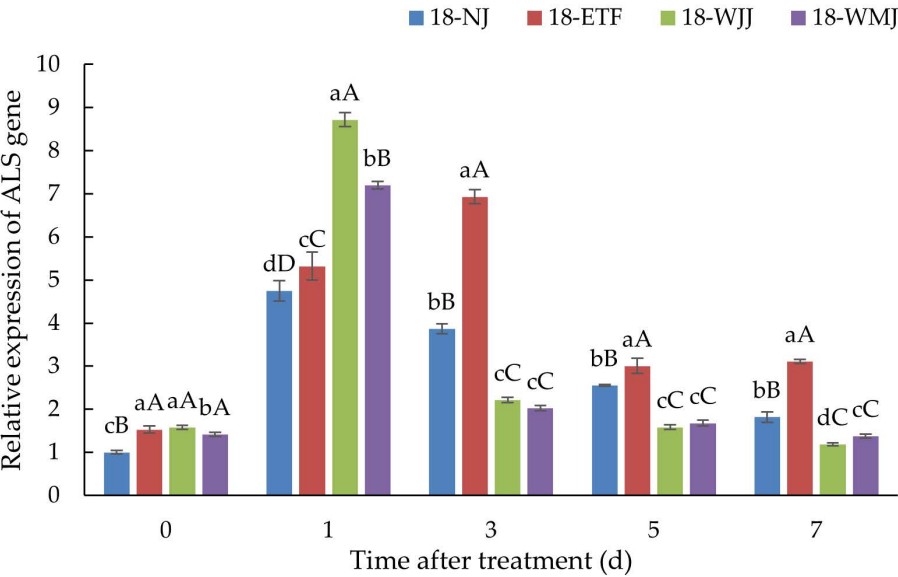

**Figure 4 Expression profiles of ALS genes in R populations (18-ETF, 18-WJJ, and 18-WMJ) and S population (18-NJ) at different penoxsulam treatment stages by RT-qPCR.** The upper-case letters represent significant differences at $p < 0.01$. The lower-case letters represent significant differences at $p < 0.05$.

levels of *ALS* in the three R populations were 1.53-, 1.58-, and 1.41-fold greater than those in the S population. However, after penoxsulam treatment, the *ALS* expression in the R and S populations significantly increased, indicating that the *ALS* gene was induced by penoxsulam. Compared with the S population 18-NJ, 18-WJJ and 18-WMJ presented the same *ALS* expression pattern. 18-WJJ and 18-WMJ exhibited the highest degree of induced *ALS* expression, with 8.72- and 7.19-fold changes at 24 h post-treatment. Subsequently, *ALS* gene expression in these two populations decreased and was lower than in the S population 18-NJ. Unlike the 18-WJJ and 18-WMJ populations, the *ALS* gene expression of 18-ETF presented another pattern, with the highest *ALS* gene expression (6.93-fold) occurring three days after treatment (Fig. 4). The different gene expression patterns may be caused by distinct mutation sites. Furthermore, NTSR may also play an important role in penoxsulam absorption and transduction, which could explain the differential expression patterns observed among populations.

## DISCUSSION

Beginning in 2008, penoxsulam has been widely and consistently utilized in China for the control of Poaceae weeds in rice fields. Consequently, an increasing number of penoxsulam-resistant populations of *Echinochloa* spp. have been identified in rice cultivation areas throughout China (*Chen et al., 2016*; *Liu et al., 2019*; *Fang et al., 2019a*; *Yang et al., 2021*; *Feng et al., 2022*; *Gao et al., 2023*). Hence, continuously monitoring resistance levels and elucidating the mechanisms of resistance to penoxsulam in barnyardgrass (*Echinochloa crus-galli*) is important for developing effective control strategies to ensure sustainable rice cultivation.

TSR has been identified as the primary mechanism responsible for conferring resistance to ALS inhibitors. This mechanism involves mutations in the target site of the ALS enzyme, which reduces the binding affinity of the herbicide, thereby rendering it less effective in inhibiting enzyme activity (*Yu & Powles, 2014*). In previous studies, nine mutation sites at Ala-122, Pro-197, Ala-205, Phe-206, Asp-376, Arg-377, Trp-574, Ser-653, and Gly-654 on the ALS protein were reported to be related to resistance to ALS inhibitors (*Fang et al., 2022*). Among these mutations, the Trp-574-Leu substitution was the most frequently reported mutation across diverse weed species (*Yu & Powles, 2014*). In barnyardgrass (*Echinochloa crus-galli* (L.) P. Beauv.) and other *Echinochloa* species, substitutions at Ala-122, Pro-197, Ala-205, Phe-206, Asp-376, Trp-574, and Ser-653 that confer resistance to ALS-inhibiting herbicides have been reported (*Panozzo et al., 2013*; *Panozzo et al., 2017*; *Riar et al., 2013*; *Matzenbacher et al., 2015*; *Liu et al., 2019*; *Fang et al., 2019a*; *Fang et al., 2022*). For *E. crus-galli*, Ala-122 mutations are the most commonly reported, followed by Trp-574 mutations, both of which play significant roles in penoxsulam resistance (*Damalas & Koutroubas, 2023*). The Trp-574-Leu mutation has also been documented in other resistant weeds, such as *Lithospermum arvense* (*Wang et al., 2019*), *Glebionis segetum* (*Papapanagiotou et al., 2023*), *Cyperus difformis* (*Huang et al., 2022*), *Poa annua* (*Vijayarajan et al., 2023*), and *Lolium multiflorum* (*Altop et al., 2022*). In our study, nine out of fifteen R populations from Hubei Province harbored the Trp-574-Leu mutation, making it the most frequent target-site mutation identified. Furthermore, a novel Pro-197-Leu substitution was identified for the first time in a population of *E. crus-galli*. Although Pro-197 mutations (where proline can be replaced by various amino acids such as Ser, His, Thr, Arg, or Leu) constitute a well-established TSR mechanism for ALS-inhibitor resistance in multiple weed species, such as *Lactuca serriola* (*Merriam et al., 2023*), *Sagittaria trifolia* (*Zou et al., 2023*), *L. rigidum* (*Kaloumenos et al., 2012*), *Schoenoplectus juncoides* (*Sada, Ikeda & Kizawa, 2013*), and *Beckmannia syzigachne* (*Wang et al., 2020*). More research will be conducted on this Pro-197-Leu biotype of *E. crus-galli*. In addition, no *ALS* mutations were detected at Ala-122, Ala-205, Phe-206, Asp-376, or Ser-653 in any of the tested populations.

To further investigate resistance mechanisms, one susceptible (S) population (18-NJ) and three resistant (R) populations (18-ETF, 18-WMJ, and 18-WJJ) were analyzed using *in vitro* ALS enzyme activity assays and *ALS* gene expression assays. Compared with the S population, all R populations presented significantly greater basal ALS activity. Upon the addition of penoxsulam to the *in vitro* assay, the ALS activity in all R populations decreased. However, this reduction was significantly less pronounced than that in the S population. Therefore, the significantly higher IC$_{50}$ of the R populations suggested reduced sensitivity of their ALS to penoxsulam. Elevated ALS activity has been documented in resistant biotypes harboring mutations such as Trp-574-Leu or Pro-197-Ser across various systems, including weeds, mutant cell lines, transgenic plants, and yeast (*Yu et al., 2010*). Our results are consistent with the presence of such target-site mutations contributing to penoxsulam resistance. In terms of *ALS* gene expression, compared with the S population 18-NJ, the R populations 18-ETF, 18-WMJ, and 18-WJJ presented significantly greater *ALS* gene expression. Although elevated gene expressions or transcript stability could lead to

increased ALS enzyme accumulation, the precise contributions of these factors to herbicide resistance remain unclear (*Yu et al., 2020*). Higher gene expression in R populations may be an important factor enhancing herbicide tolerance, but additional data are needed to validate this hypothesis. Importantly, transcription levels do not always directly correlate with translation processes, and post-translational regulation could also influence actual protein levels (*Kuhlemier, 1992*).

Furthermore, *E. crus-galli* is a hexaploid species that evolved through hybridization of the tetraploid *E. oryzicola* Vasinger and an unknown diploid species (*Aoki & Yamaguchi, 2008*) and harbors three distinct *ALS* gene copies (*Panozzo et al., 2021*). This complex genomic architecture contributes significantly to intricate herbicide resistance mechanisms. Research has shown that any of the *ALS* gene copies can harbor ALS inhibitor resistance mutations (*Panozzo et al., 2021*), which may also explain why these weeds are prone to developing resistance to ALS inhibitors. In hexaploid species such as *E. crus-galli*, the presence of multiple homoalleles across subgenomes creates a genetic environment in which resistance mutations can arise and quickly become prevalent. This rapid enrichment of resistance genes is facilitated by the high initial frequency of resistance genes, the dominant genetic inheritance patterns, and the fitness costs associated with the resistance alleles. More importantly, the expression level of the mutated *ALS* gene is strongly linked to herbicide resistance. For example, in *Monochoria vaginalis* (which possesses four functional *ALS* genes), resistance mutations were identified only in the two most highly transcribed *ALS* genes (*Tanigaki et al., 2021*). This finding suggests that mutations in genes with low expression may not confer sufficient resistance. Therefore, future research will focus on determining which specific *ALS* gene copy within *E. crus-galli* harbors the resistance mutation(s) and how these mutations influence the expression levels of the respective *ALS* genes.

NTSR mainly encompasses mechanisms that involve the metabolism of herbicides or the redirection of herbicides away from their intended sites of action. In particular, the NTSR mechanism may include the detoxification of herbicides by plant endogenous enzymes such as glutathione S-transferases (GSTs), glycosyltransferases (GTs), cytochrome P450s, and ATP-binding cassette (ABC) transporters. Furthermore, the NTSR mechanism has been documented in numerous cases of herbicide-resistant weeds, such as *E. crus-galli* (*Fang et al., 2019b*), *L. rigidum* (*Han et al., 2021*), *Descurainia sophia* (*Shen et al., 2022*), and *B. syzigachn* (*Bai et al., 2022*). Some NTSR-related genes have been identified, and their functions have been confirmed. In the case of late watergrass (*E. phyllopogon* (Stapf) Koso-Pol.), two homologous cytochrome P450 genes, *CYP81A12* and *CYP81A21*, have been identified as conferring resistance to the ALS inhibitors bensulfuron-methyl and penoxsulam (*Iwakami et al., 2014*). Another *CYP81A68* confers metabolic resistance to ALS-inhibiting herbicides in *E. crus-galli* (*Pan et al., 2022*). Aldo-keto reductase plays a vital role in *E. colona* resistance to glyphosate (*Pan et al., 2019*). NTSR-related genes have also been identified in other weed species. The cytochrome P450 *CYP709C56* metabolizing mesosulfuron-methyl confers herbicide resistance in *Alopecurus aequalis* (*Zhao et al., 2022*). The P450 gene *CYP749A16* is required for tolerance to the sulfonylurea herbicide trifloxysulfuron sodium in cotton (*Gossypium hirsutum* L.) (*Thyssen et al., 2018*). The

cytochrome P450s *BsCYP99A44* and *BsCYP704A177* confer metabolic resistance to ALS herbicides in *B. syzigachne* (*Bai et al., 2022*). ABCC8 confers glyphosate resistance in plants (*Amrhein & Martinoia, 2021*; *Pan et al., 2021*). Therefore, further studies should be conducted to confirm the possible resistance mechanisms involved in the four penoxsulam-resistant *E. crus-galli* populations without any target site mutations.

## CONCLUSIONS

In conclusion, fifteen penoxsulam-resistant *E. crus-galli* populations (18-ETF, 18-DDX, 18-WLH, 18-WMJ, 18-WJJ, 18-QZW, 18-JPX, 18-JJCY, 18-YTW, 18-HDD, 18-XYCZ, 18-JWZ, 18-JZWD, 18-XYXS, and 18-JJXJ) were used to explore resistance mechanisms. Eleven populations (18-ETF, 18-DDX, 18-WLH, 18-WMJ, 18-WJJ, 18-QZW, 18-JPX, 18-JJCY, 18-YTW, 18-HDD, and 18-XYCZ) exhibited mutations of the target enzyme ALS. Overexpression of the *ALS* gene and improvement in ALS enzyme activity also contributed to resistance in 18-ETF, 18-WMJ, and 18-WJJ populations. The results showed that the altered target site is a mechanism of resistance to penoxsulam in 18-ETF, 18-WMJ, and 18-WJJ populations. The remaining four populations (18-JWZ, 18-JZWD, 18-XYXS, and 18-JJXJ) had no target site mutations of the *ALS* gene, and further research is needed to determine their possible resistance mechanism. Comprehensive analysis of the resistance level of barnyardgrass to penoxsulam in Hubei rice fields, coupled with a deep understanding of the underlying resistance mechanisms, is crucial. This knowledge enables the development and implementation of targeted resistance management strategies. Such strategies will effectively slow the development and spread of resistance, promote rational herbicide application practices, and ultimately contribute to the long-term ecological and economic sustainability of rice production in Hubei Province.

## ACKNOWLEDGEMENTS

We thank Prof. Yongfeng Li from the Institute of Plant Protection, Jiangsu Academy of Agricultural Sciences, for providing the sensitive populations of *E. crus-galli*. Thanks are also due to the reviewers and editors for their helpful comments on earlier drafts of the manuscript.

### Funding

This work was supported by the Youth Science Foundation of Hubei Academy of Agricultural Sciences (2020NKYJJ09), the Hubei Provincial Innovation Center for Agricultural Sciences and Technologies (2021-620-000-001-013) and the Youth Science Foundation of Institute of Plant Protection and Soil Fertilizer, Hubei Academy of Agricultural Sciences (2021ZTSQJ04). The funders had no role in study design, data collection and analysis, decision to publish, or preparation of the manuscript.

## Grant Disclosures

The following grant information was disclosed by the authors:

The Youth Science Foundation of Hubei Academy of Agricultural Sciences: 2020NKYJJ09.

The Hubei Provincial Innovation Center for Agricultural Sciences and Technologies: 2021-620-000-001-013.

The Youth Science Foundation of Institute of Plant Protection and Soil Fertilizer, Hubei Academy of Agricultural Sciences: 2021ZTSQJ04.

## Competing Interests

The authors declare there are no competing interests.

## Author Contributions

- Qiongnan Gu conceived and designed the experiments, performed the experiments, analyzed the data, prepared figures and/or tables, authored or reviewed drafts of the article, and approved the final draft.
- Jing Shen performed the experiments, authored or reviewed drafts of the article, and approved the final draft.
- Shihai Chu conceived and designed the experiments, authored or reviewed drafts of the article, and approved the final draft.
- Qichao Huang performed the experiments, analyzed the data, prepared figures and/or tables, and approved the final draft.
- Anan Chen performed the experiments, prepared figures and/or tables, and approved the final draft.
- Lin Li performed the experiments, prepared figures and/or tables, and approved the final draft.
- Ruhai Li conceived and designed the experiments, authored or reviewed drafts of the article, and approved the final draft.

## Field Study Permissions

The following information was supplied relating to field study approvals (*i.e.*, approving body and any reference numbers):

Filed experiments were approved by the research council of the Institution of Plant Protection and Soil Fertilizer, Hubei Academy of Agricultural Sciences.

## DNA Deposition

The following information was supplied regarding the deposition of DNA sequences:

The ALS sequences with mutation sites are available at GenBank: PV393064 to PV393073, PV405280.

## Data Availability

The raw data are available in the Supplemental Files.

## Supplemental Information

Supplemental information for this article can be found online at http://dx.doi.org/10.7717/peerj.19973#supplemental-information.

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
