# Peer review of "Analysis of the resistance level and target site resistance mechanisms of Echinochloa crus-galli to penoxsulam from Hubei Province, China"

_PeerJ, doi:10.7717/peerj.19973_

## Round 0.1 · original submission · Minor Revisions

Herbicides are among the most common agrochemicals in conventional production systems. Their excessive use may cause problems in agricultural fields, including herbicide-resistant weeds. Therefore, I believe any study to enhance our understanding of herbicide resistance and its mechanisms is highly valuable. However, it is essential to address certain technical details to enhance the article further. I strongly recommend carefully reviewing the reviewers' suggestions and thoughtfully considering each recommendation. If you disagree with any suggestion, it would be helpful to provide clear, well-reasoned justifications for your viewpoint.

**Language Note:** The review process has identified that the English language must be improved. PeerJ can provide language editing services - please contact us at [email protected] for pricing (be sure to provide your manuscript number and title). Alternatively, you should make your own arrangements to improve the language quality and provide details in your response letter. – PeerJ Staff

Reviewer 1 ·

Basic reporting

Regarding the manuscript ID-112509, the authors have demonstrated that Echinochloa crus-galli populations from Hubei, China fields have developed significant resistances to penoxsulam, an ALS-inhibiting herbicide, and resistant mechanism. The manuscript is written in clear and grammatically correct English and provides interesting results and discussion. However, some sections should be improved and clarified.

Experimental design

- Introduction provides a general overview of barnyardgrass and its chemical control, however, it lacks a clear statement of gap and hypothesis of the study. Paragraphs 3 and 4 of Introduction part must be rearranged. Penoxsulam should be introduced before possible herbicide resistance mechanism. Lines 56-57: does the author mention target-site mechanism cases of E. crus-galli to penoxsulam? If so, please state.

- Materials and Methods should be clarified/corrected:
• Did the authors collect the seeds of E. crus-galli from rice fields?
• What is the recommended rate of penoxsulam in the field?
• How many barnyardgrass populations were used in the study?
• Please specify the low temperature or room temperature in Lines 86, 87, 94.
• Were the studies of whole-plant dose response and ALS activity assay repeated?
• There was no report of the herbicide use records of the fields where a penoxsulam resistant populations were collected. In addition, the herbicide use records can help in discussion about selection pressure and resistance level.
• Please state the length of ALS gene. Did the authors get a full length or a partial sequencing of ALS gene from a pair of primer? Please provide the conserved domains of ALS gene based on the used primers.
• Did the authors modify the ALS activity assay from Yu et al. (2010)? Please briefly describe the assay used in the study.
• Line 168: What was comparison mean by the lowest standard deviation (LSD)? Was it “The Least Significant Difference (LSD)”?
• Did the authors perform the lack-of-fit F test on the nonlinear log-logistic regression model? This test can help to verify that the GR50 or IC50 was accurate.

Validity of the findings

The findings is meaningful. However, some parts should be explained and clarified.
- Based on Lines 106-107, the penoxsulam was applied at rates ranged from 4, 8, 16, 32, 64, 128, 256, 512 g a.i./ha on the putative resistant populations. However, in Table 2 the GR50 of 18-ETF at 21 days after application was reported at 750.75 g a.i./ha. The estimated GR50 was far beyond the application rates. In addition, the GR50 of 18-XYXS and 18-JJXJ were 3.52 and 2.88, respectively, which were lower than the application rates of penoxsulam. Therefore, there were insufficient data points to estimate the accuracy of GR50 and resistance index calculation was inaccurate for 18-ETF, 18-XYXS and 18-JJXJ populations.

- Line 184: There were 58 E.crus-galli populations collected. However, Tables and Figures and the rest of the paper mentioned only 15 E.crus-galli populations.

- Line 242-243: Please explain why there were different ALS gene expression patterns among resistant populations.

- Based solely on sequencing data, the authors conclude that four populations exhibit non-target-site resistance (NTSR) mechanisms, as shown in the results and Table 3. I suggest the authors avoid overestimating these findings, as there is no supporting physiological or biochemical evidence.

Additional comments

- Abstract should be improved to present findings clearly and concisely. Lines 26-28: reported the higher ALS activity, higher doses of penoxsulam, and higher ALS gene expression without clearly state. Please explain how high these parameters.

More general comments:
Please italicize “in vivo” “in vitro” throughout the document.
Please space before the parenthesis include Lines 39, 221, 322 and the rest.
Please italicize “ALS/AHAS” gene throughout the document.

Annotated reviews are not available for download in order to protect the identity of reviewers who chose to remain anonymous.

·

Basic reporting

The authors indicate that the object of the research is ALS-resistant Echinochloa crus-galli. However, there are at least 8 species of herbicide-resistant Echinochloa known worldwide, and these species are often difficult to identify. It is desirable to provide confirmation that we are dealing with Echinochloa crus-galli.
The authors point out that there is resistance in Echinochloa crus-galli to herbicides with 6 different mechanisms of action. However, 7 are known to date:
Inhibition of Acetyl CoA Carboxylase HRAC Group 1;
Inhibition of Acetolactate Synthase HRAC Group 2;
Inhibition of Microtubule AssemblyHRAC Group 3;
Auxin Mimics HRAC Group 4;
Inhibition of Enolpyruvyl Shikimate Phosphate Synthase HRAC Group 9;
Very Long-Chain Fatty Acid Synthesis inhibitorsHRAC Group 15;
Inhibition of Cellulose Synthesis HRAC Group 29;
Also, 29 mutations of the action site are known to lead to ALS-resistance of Echinochloa crus-galli var. crus-galli (line 48-67).
Indicate the BBCH phase of plant development in the text
Line 143: indicate the number of analytical repeats.

Experimental design

no comment

Validity of the findings

no comment

Additional comments

no comment

Reviewer 3 ·

Basic reporting

Summary
This manuscript investigates resistance to penoxsulam in Echinochloa crus-galli populations from Hubei Province, China. The detection of the Trp-574-Leu mutation in nine populations and the variation in ALS gene copies (ALS1, ALS2, ALS3) carrying the mutation is both novel and relevant. Additionally, combined expression analysis of ALS gene copies in selected populations adds valuable insight into possible resistance enhancement through overexpression.

Major Comments
Mutation Reporting: The Pro-197-Leu mutation is mentioned in the text but not shown in Table 3. Please confirm whether this mutation was detected and update the table if applicable, or clarify if it was not observed in any of the tested populations.

Dose–Response Curves: While GR₅₀ values are reported, inclusion of dose–response curves for selected populations (e.g., representative resistant and susceptible biotypes) would improve data visualization and help readers better understand the degree and variability of resistance.

ALS Gene Expression: The manuscript appropriately states that ALS1, ALS2, and ALS3 were amplified collectively for gene expression analysis. However, it would still be helpful to:

Indicate how “high expression” was quantified (e.g., fold-change compared to the susceptible population).

Mutation in Different ALS Copies: The finding that the same mutation (e.g., Trp-574-Leu) occurs in different ALS gene copies across populations is intriguing. A brief discussion on why different gene copies may acquire mutations — and how that might affect resistance development or management — would be a valuable addition.

Minor Comments
The manuscript is clearly written with minimal language issues.

Geographic context and population mapping are well presented.

Recommendation
Minor Revision – This is a well-conducted and timely study. Addressing the mutation table discrepancy, enhancing visual data with selected dose–response curves, and expanding on the implications of gene copy-specific mutations will further strengthen the manuscript.

Experimental design

The experimental design is well thought out and clearly presented in the manuscript.

Validity of the findings

The experimental design is solid and well described. The methodology is appropriate for addressing the research objectives, including herbicide dose–response assays, ALS gene sequencing, and expression analysis. The data support the conclusions, and there are no major flaws in experimental execution or interpretation. The detection of Trp-574-Leu mutations and gene expression patterns are valid and relevant to the study of ALS inhibitor resistance.

Additional comments

none

---

## Round 0.2 · Minor Revisions

I would like to thank you for accepting the reviewers’ suggestions and improving your article accordingly. Your manuscript will be ready for publication after a language revision. Below are some comments from our section editor regarding the language. I recommend seeking assistance from a colleague or using our editing service to ensure the language is clear and professionally polished.

"The grammar and writing is not acceptable for publication. Even in conclusion, most of the sentences are wrong: "Of those, eleven populations endowed with TSR mechanism, which exists target enzyme ALS mutation. And the overexpression of ALS gene and the improvement of ALS enzyme activity also contributed resistance in some resistant populations..." your statements also should be specific and accurate ("some resistant populations" => how many/which %?). In addition: Provide an explanation of the resistance mechanism and possible strategies to deal with it."

·

Basic reporting

no comment

Experimental design

no comment

Validity of the findings

no comment

Reviewer 3 ·

Basic reporting

The authors have addressed the key research questions effectively. The study is well designed, and the data presented support the conclusions. The manuscript provides clear answers to the central objectives regarding resistance levels and ALS-target site mechanisms in Echinochloa crus-galli.

Experimental design

The experimental design is appropriate and well-structured for the objectives of the study. The authors used a clear workflow involving population collection, dose–response assays, target-site sequencing, and gene expression analysis. Replication and controls were adequately included, and the methodology is sufficiently detailed to ensure reproducibility. The design allows for confident interpretation of both phenotypic resistance levels and underlying molecular mechanisms.

Validity of the findings

The findings are valid and well supported by the data. The dose–response results clearly distinguish resistant from susceptible populations, and the identification of ALS target-site mutations (especially Trp-574-Leu) is consistent with known resistance mechanisms. The expression analysis further reinforces the conclusions in selected populations. Overall, the methodology is sound, and the interpretation of the results is appropriate and aligned with the objectives of the study. No major flaws were identified in data collection, analysis, or reporting.

Additional comments

None

---

## Round 0.3 · accepted · Accept

I would like to thank you for accepting the referees' suggestions and improving your article based on their suggestions. Your article is ready to publish. We look forward to your next article.